# Land Competition under Telecoupling: Distant Actors' Environmental versus Economic Claims on Land in North-Eastern Madagascar

**O. Ravaka Andriamihaja** [1,2,*] , **Florence Metz** [3,4], **Julie G. Zaehringer** [1] , **Manuel Fischer** [3,5] **and Peter Messerli** [1,2]

1    Centre for Development and Environment, University of Bern, Mittelstrasse 43, CH-3012 Bern, Switzerland; julie.zaehringer@cde.unibe.ch (J.G.Z.); peter.messerli@cde.unibe.ch (P.M.)
2    Institute of Geography, University of Bern, Hallerstrasse 12, CH-3012 Bern, Switzerland
3    Institute of Political Science, University of Bern, Fabrikstrasse 8, CH-3012 Bern, Switzerland; florence.metz@ipw.unibe.ch (F.M.); Manuel.Fischer@eawag.ch (M.F.)
4    Natural Resource Policy Group, Institute of Environmental Decisions, ETH Zürich, Universitätstrasse 22, CH-8092 Zürich, Switzerland
5    Eawag (Swiss Federal Institute of Aquatic Science & Technology), Überlandstrasse 133, CH-8600 Dübendorf, Switzerland
*    Correspondence: ravaka.andriamihaja@cde.unibe.ch

**Abstract:** In north-eastern Madagascar, maintenance of biodiversity competes with expansion of land for agriculture and mining. The concept of "telecoupling" provides a framework for analysis of distant actors and institutions that influence local land use decisions. However, there is a lack of knowledge regarding the extent of telecoupling of land governance in north-eastern Madagascar and a lack of evidence regarding its role in driving land use change and land competition. Using a descriptive Social Network Analysis, we disentangled distant interactions between actors in terms of flows and institutions. Our findings show that the domains of economic and environmental interactions are dominated by actors from different sectors that have claims on the same land but generally do not interact. Distant influences occurring via remote flows of goods, money, and institutions serve to reinforce local land competition. Balancing economic and environmental land claims for more sustainable regional development in north-eastern Madagascar requires collaboration between actors across sectors, scales, and domains.

**Keywords:** Land system science; land governance; land competition; telecoupling; Social Network Analysis; Madagascar

## 1. Introduction

Global resource consumption continues to rise, particularly affecting tropical countries [1,2], where changes in land use reflect a drive towards large-scale commercial agricultural [3,4]. These land use changes are significantly shaped by a growing number of diverse connections between local and distant actors. The phenomenon of distant connections is conceptualized as "telecoupling", which is described as "a newly developed umbrella concept that encompasses a broad range of socioeconomic and environmental interactions over distances" [5]. In this way, distant locations (e.g., European food-processing firms) are socioeconomically and environmentally interconnected with local sites in the global South, and vice versa. Various remote actors can affect local land use change and the governance thereof. Liu et al. [6] illustrate such distant interactions, showing how the soy trade between Brazil and China, for example, led to agricultural land use intensification in Brazil. At the

same time, China experienced an increase in afforestation and concomitant carbon sequestration. In a later study, however, Sun et al. [7] underlined the negative environmental consequences of this soy trade in China. The low price of the imported soy, which Chinese soy farmers could not compete with, pushed them into replacing their soy plantations with more nitrogen-demanding crops (such as wheat, corn, and rice), causing nitrogen pollution. The expansion of commercial agricultural plantations in the global South affects land use in these regions as well as elsewhere, from the range of tropical forests to subsistence crop fields [8]. Further, these distant connections impact the economic development of the tropical countries in question [9], while potentially undermining other globally driven initiatives such as conservation [10].

Telecoupling can lead to local land competition, as different (local and distant) actors demand goods or services from the same finite land [11]. A well-known example is the trade-off between food crops versus biofuel crops, a type of land use competition that can be described as "production vs. production" [12]. This type of land competition can affect food prices and supplies [13]. Another example is the privatization of agriculture, such as in Nigeria, which often crowds out smallholder production systems. The latter, in turn, must shift and expand elsewhere, thereby threatening conservation initiatives and biodiversity-rich forests, as we see in Nigeria [14]. Such restrictions of access to land for local populations, and resulting movement of small-scale land use, also take place when conservation actors implement protected areas [15–17]. To mitigate these dynamics and improve the performance of protected areas, "Integrated Conservation and Development Projects" (ICDPs) were introduced, especially in the 1990s, aiming to link and mutually foster conservation and development. However, in several cases, these projects have failed to conserve biodiversity and have not brought development to local populations; examples include the Kakamega forest project in Kenya [18] as well as the Ranomafana, Andohahela, and Masoala National Parks in Madagascar [19–21]. Studies indicate that a lack of attention to contextual socioeconomic factors [18] and insufficient compensation of the local population [20] contributed to these failures.

Land system science [22,23] uses the concept of land governance to understand the drivers of land system change as well as to design sustainable transformation actions [23]. Graham et al. [24] define governance as "the interactions among institutions, processes, and actors that determine how power and responsibilities are distributed, how decisions are taken, and how citizens or other stakeholders are able to have their say". In this way, land governance is the result of both formally stated rules and informal interactions between actors [25,26] carrying out land-related activities. By integrating the concept of telecoupling, land governance studies benefit from a shift in focus from territory governance to flow-centered governance [27]. Long-distance flows of commodities, capital, and people dominate land systems beyond clearly defined territories [28]. Moreover, local and national actors partly derive from and share their authority with transnational and distant actors that contribute knowledge, advice, and resources [29]. Transnational actors can involve public actors and private companies. Indeed, in addition to state authorities, private sector actors (e.g., transnational investors and firms) often have considerable influence on land governance [27,30,31]. Consequently, telecoupled land governance typically involves cross-border interactions between public and private organizations embedded in complex network structures [32–34].

Social Network Analysis (SNA) provides a means to disentangle these interactions and operationalize the telecoupling framework [35]. Land science scholars have applied network approaches to study transnational [36] and telecoupled [6] systems. Network analysis can capture the diversity of actors involved in land governance and their various types of interactions in terms of flows (of money, knowledge, goods, etc.) and shared institutions. Bodin and Crona [37] point to the importance of understanding network characteristics to comprehend social processes. The decisions emerging from interactions between numerous different local and distant actors are crucial for land systems and the dynamic interactions between socio-economic and biophysical factors [38]. However, better understanding of land governance networks in complex telecoupled situations is needed for robust empirical land governance and land policy [39,40].

In this paper, we focus on Madagascar, a global biodiversity hotspot in the Indian Ocean that has been subject to various land claims by distant actors since colonial times [41,42]. Due to Madagascar's high degree of endemic plant and animal species and the related high-profile interests of Western conservationists, the island saw its system of protected areas expand rapidly in the last two decades [43]. Despite claims of resulting global benefits [20], this "conservation rush" did not actually halt deforestation or biodiversity loss on the island, nor did it alleviate poverty [17,20,43,44]. Indeed, the creation of these protected areas competes with the livelihoods of local communities [17,43,45]. It remains an enormous challenge to reconcile the claims of different actors on the environmental and economic functions of land in Madagascar.

To improve understanding of the situation, this paper analyzes the extent of telecoupling regarding land governance in north-eastern Madagascar, and its role in driving land use change and land competition via actor networks. We empirically investigate governance networks with respect to two key land use dynamics [12]: production and conservation. In particular, our study seeks to answer the following research question: "Does telecoupled land governance foster land competition in north-eastern Madagascar?" We begin by describing land competition under telecoupling, and then analyze the distant influences in terms of flows and institutions.

## 2. Study Areas

For the present research, we adopted a case study approach, investigating actor networks connected to local land use changes from the bottom-up. The landscapes in our case study are located in the district of Maroantsetra, within the Analanjirofo region of north-eastern Madagascar. This region is home to an important part of Madagascar's remaining humid forest, which is surrounded by a highly diverse landscape mosaic scattered with small-scale agriculture [46]. More specifically, our two case study landscapes each encompass two villages located in two different administrative communes: the villages of Morafeno and Beanana in the Morafeno commune; and the villages of Mahalevona and Fizono in the Mahalevona commune (Figure 1).

Both our case study landscapes have undergone a variety of significant land use changes, in particular transitions from shifting cultivation of subsistence rice to permanent cultivation of clove and vanilla in agroforests, as well as from unprotected to protected forest. Firstly, our Morafeno case study landscape borders the Makira Natural Park (est. 2012), a REDD+ project (Reducing Emissions from Deforestation and Forest Degradation) since 2005. Secondly, our Mahalevona case study landscape borders the Masoala National Park (est. 1997) [19,45]. These protected areas enclose an important part of the island's remaining biodiversity-rich humid tropical forest, and are meant to ensure the continuing provision of ecosystem services [43]. With the establishment of the parks, forest clearing was made illegal within their boundaries. However, forest cutting is still widely practiced by the local population. The same is true of artisanal mining in and around the protected areas. As a result, community-based natural resource management (CBNRM) initiatives were set up in the buffer zones of the protected areas, in an effort to conserve the remaining forest fragments they encompass.

The Analanjirofo region benefits from an ideal climate and soil conditions for cultivation of various cash crops, such as clove and vanilla, which were introduced during colonial times (1894–1960) and mainly produced for export [47]. Indeed, Analanjirofo means "forest of cloves" in Malagasy. Farmers in the region appreciate clove and vanilla crops due to their increasing market price and historic value as export products. For many years, clove made up the largest proportion of Madagascar's agricultural exports [48]. Today, vanilla occupies the top spot [49]. The recent "vanilla boom" has enabled an enormous influx of cash to some local farmers [50], further increasing pressure on remaining forests. To produce the main Malagasy staple crop of rice for subsistence, local small-scale farmers practice the traditional, highly adapted land use system of shifting (or swidden) cultivation. However, as in many other tropical regions around the world, external actors have long deemed shifting cultivation to be a "backwards" practice of uneducated rural people, also blaming it for deforestation. Nevertheless, attempts to halt the expansion of shifting cultivation landscapes via policies, enforcement, or fostering

of agricultural intensification have not shown much success to date [46]. In addition to practicing shifting cultivation, those local farmers who possess suitable flat/irrigable land often grow rice in irrigated paddy fields [51]. Further, some villagers keep zebu cattle that graze on the few local pastures or on the irrigated rice fields after harvest. Particularly in the Morafeno case study landscape, local land users also practice artisanal quartz mining in fields, along the rivers, and in the forests surrounding the villages. In the case of Mahalevona village, silkworm breeding enables some farmers to diversify their income.

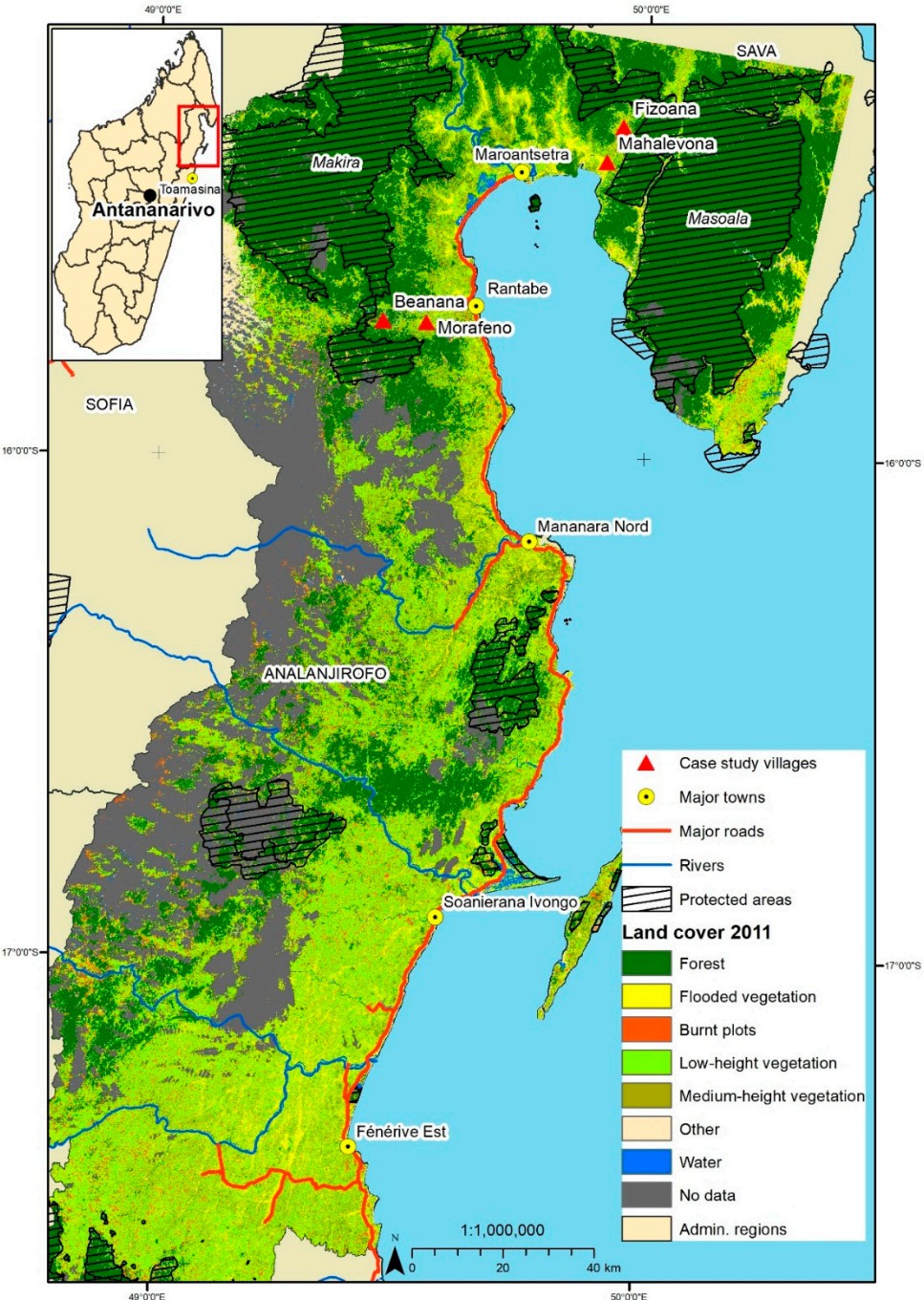

**Figure 1.** Location of study area in the Analanjirofo region of north-eastern Madagascar.

Even today, the Analanjirofo region remains extremely remote and difficult to access, and displays a very low level of infrastructure. Villages located in the hinterlands are generally only accessible by foot, and lack basic sanitation and phone networks. Electricity has only become available in the last

three years, thanks to small Chinese solar panels. Further, the region is frequently hit by cyclones that destroy settlements, cash crops, and irrigated rice fields. Cash crop prices have varied greatly over the years, such that few local farmers have been able to escape poverty over the long term. Indeed, many households in our case study landscape still suffer from an annual "hunger gap".

This biophysical and socioeconomic mismatch—between local resource wealth and ongoing human poverty—points to the need for better understanding of the role of land governance regarding the social–ecological systems in place. Conservation funding [43], on the one hand, and the global trade of cash crops [50], on the other, link the local social-ecological systems with the rest of the world, rendering comprehensive analysis of area land governance highly complex. Nevertheless, the region provides a prime case for examination of land competition between the interests of conservation and agricultural development [47]. In the next section, we will explain how we operationalized the telecoupling concept to shed a light on the local implications of global decisions.

## 3. Materials and Methods

### 3.1. Data Collection

#### 3.1.1. Focus Groups per Village

Between April and November 2016, we conducted preliminary focus group interviews in each of the two study villages, in preparation for further survey data collection. We established focus groups with the twofold aim of obtaining an overview of (a) the main land uses and land use changes (hereinafter LU and LUC); and (b) the main organizational actors present in each of the study villages. These two elements provided the starting point for our expanded analysis. Together with our local partners, we invited a variety of actors representing diverse LU activities—such as CBNRM, rice and vanilla cultivation, cash crop plantations, and artisanal mining—to become members of our focus groups. Those actors who accepted our invitation include local associations, local authorities, individual farmers, and traders.

In order to prepare the survey, we proceeded in three steps: First, we asked 5–10 members of each focus group to identify and take note of the main LU and LUC that occurred in their respective villages within the last 20 years. Secondly, members of focus groups listed the actors influencing or being affected by each LU and LUC, as well as the factors causing the changes. In the process of identifying factors causing the LUC, focus group members mentioned additional actors related to LUC in the villages. Thirdly, we compiled an exhaustive list of actors named by the focus groups as being involved in LU and LUC for each of the villages. Finally, we used these lists of actors as a starting point for actor surveys.

#### 3.1.2. Survey Population

*Actors*, as defined here, include groups and organizations who are active in land governance or who derive their means and meanings from land [52]. On the village level, we consider actors as categories of individuals (e.g., vanilla farmers, traders) practicing the same activity and broadly pursuing the same land-related aims at the local level. Beyond the village level, we consider actors as organizations (e.g., national agencies), thereby building on established definitions of organizational actors [53,54]. We originally developed the survey in English and then translated it into French and Malagasy. Depending on the main language of the respondent, we used one of these three versions. The survey was pretested in the village of Mahalevona in November 2016 with five groups/individuals: farmers, collectors of cash crops, Tangalamena or elders, a representative of the water management association, and the chief of the village of Mahalevona.

In order to identify further actors relevant to LU and LUC in the case study villages and beyond, we applied snowball sampling. The snowballing method involves identifying subsequent sets of respondents from those that precede them. In our case, the initial respondents were those identified in

our focus groups. During the survey of these actors, we identified further actors with whom this initial actor set exchanged goods and money, or who adhered to the same institutions. Next, we surveyed the newly identified actors, and repeated the snowballing procedure. Respondents were often vague when they spoke about organizational actors they exchanged goods or money with. Rather than mentioning the name of an organization, respondents often referred to broad categories of actors (e.g., "intermediary" for a commodity gatherer using the money of a "collector"). In these cases, we searched for the organization that best represented the described actor group.

At the same time, we applied a threefold set of criteria to narrow down the relevant survey population, rather than simply surveying every actor mentioned in the snowballing procedure. First, we identified actors involved in key flows, such as flows of conservation funding, distinguishing those directly or indirectly linked to LUC that impact human well-being and the environment in the district of Maroantsetra. Secondly, we identified actors that change the path of the flow of goods or money, such as intermediaries. Changing the path of a flow refers to actors involved in selling, processing, buying, sharing, regulating, or giving goods/money to another actor. Thirdly, we identified institutions that influence the strategy or means of actors and, thereby, influence their activities.

In this way, we sought to survey all actors named in the snowballing approach that fulfilled our threefold set of criteria. In total, we identified 143 actors relevant to the two villages—some were linked only to one village and others were linked to both villages. Specifically, 101 actors were linked to Morafeno village, and 102 actors were linked to Mahalevona village. In addition, there were a number of actors (n = 101 relevant to the two villages: n = 72 Morafeno; n = 79 Mahalevona) who fulfilled our criteria of relevance, but were either unwilling or unable to be surveyed. To fill these data gaps, we conducted expert interviews and collected secondary data. Our expert interviews—mainly concerning the trade of vanilla and clove—enabled us to gain information about the networks that non-respondents maintain to exchange goods and money as well as the institutions they adhere to. Further, we compiled secondary data from the websites of multinational companies, organizations, and Malagasy government ministries between August and October 2017, in order to obtain information about interactions among non-respondents and institutions followed.

### 3.1.3. Survey Mode and Procedure

We conducted the survey in three different ways: (1) face to face with those actors who could meet in person; (2) via Skype with those who could not meet personally, but were willing and able to be surveyed this way; and (3) via an online survey with those who could not meet personally and explicitly requested a written web/electronic survey platform (Table 1). In total, we conducted 42 surveys out of the 143 actors we originally identified as relevant.

**Table 1.** Overview of survey population in the two villages.

| Survey Mode | 34 Face-to-Face Interviews | 1 Skype Interview | 7 Online Surveys |
|---|---|---|---|
| Level of actors | 20 village actors, 11 district actors, 1 regional actor, 2 national actors | 1 international actor | 1 district actor, 4 national actors, 2 international actors |

In the field, we gathered data in three phases, beginning at the local and district levels and then moving up to higher levels in consecutive phases. Hereinafter, we define the village level as the area within the case study villages of Morafeno and Mahalevona villages. The district level refers to the area within the district of Maroantsetra. The regional level refers to the area within the region of Analanjirofo. The national level refers to the area within the Republic of Madagascar. Finally, the international level refers to all areas outside of the Republic of Madagascar. Actors were assigned to different levels based on their area(s) of activity.

The first phase, mostly consisting of face-to-face surveys (n = 20), took place from November to December 2016 at the village level. After the villages, by following the flows of goods and money, we surveyed relevant actors located at the district level, in the communes nearby our study villages and

in the district of Maroantsetra. In the second phase in January 2017, we surveyed actors at the regional and national level. Our sample included a regional actor located in Fénérive-Est and three national actors located in Toamasina and Antananarivo. In the third phase, between February and August 2017, we surveyed those national-level actors who could not be interviewed in person in Madagascar as well as the international-level actors taking part in online surveys (n = 7) or Skype interviews (n = 1). For the online survey, we adapted our questionnaire to the "lime survey" format. The international actors were located in Switzerland, the US, India, Germany, and France.

### 3.1.4. Survey Data

We distinguished actors according to the following categories:

- Sectors: private, public, voluntary ("Voluntary" refers to a diverse array of non-profit and non-governmental organizations, emphasizing civil society.) sector;
- Levels: village, district, regional, national, international level;
- Domains: economic (activity that expands commodities production/extraction), environmental (activity (Could be money-making that preserves environment) that preserves and maintains forest).

*Flows* constitute exchanges of goods and money between actors. According to Sikor et al. and Lambin et al. [27,28], an open land system consists of large flows of goods, people, and capital that connect local land users with global-scale actors. Based on this definition, we employed the flow-centred land governance concept to capture telecoupled land systems, connecting distant actors to local ones via their exchanges of material and financial flows. Accordingly, distant actors comprise all those who influence a village from the outside, while the term "outside" refers to the district, regional, national, or international levels. Goods include all merchandise and material products such as crops or agricultural inputs. Money refers to the exchange of financial capital and includes conservation or development funding or payments in exchange for goods. In the survey, respondents were asked to indicate what item(s) they exchanged with other actors (goods, money), to specify the actors they exchange the item(s) with, and to indicate the direction of that exchange (e.g., received from another actor or provided to another actor). The original question in the survey was: "Related to your activities in Maroantsetra, we would like to understand how you interact with other organizations. These interactions can comprise the exchange of items such as goods, financial capital, human resources or information. Please indicate the item you exchanged. If you received it, from whom? If you provided it, to whom? (name and contact)." A table with four columns informed us about the exchanged items, the origin (from whom?), the destination (to whom?), and any remark when needed. The analysis in this paper focuses on exchange of goods and money only.

We follow the definition of *institutions* as rules governing the behavior of actors [32,55,56]. Institutions can be formal in the form of written policies, laws, decrees, or land tenure rules; or they can be informal in the form of unwritten norms, rules, or customary rights. Formal and informal institutions influence the flow of goods and money by governing behavior. We refer to "institutional linkages" or a "shared institution" when a pair of actors adhere to the same institution when exchanging goods or money. In the survey, actors were asked to indicate LU-related rules and regulations they were aware of, and whether they adhered to them or not. The original question in the survey was: "We consider that different rules and regulations influence the strategy of your organization. These rules and regulations can be formal, such as laws, decrees, etc. but also informal such as traditions, customary rules, informal agreement or special relations to other actors. Which rules and regulations does your organization follow? Please indicate the rules and regulations your organization knows about. Then check whether you adhere it or not." A table with three columns informed us about the known institutions, the institutions adhered to (yes/no for the known institutions), and any remark where applicable.

### 3.2. Data Analysis

Our data capture the exchange of flows among actors, thus constituting relational data. Relational data are usefully analyzed by conceptualizing them as networks of nodes and ties [37,57–59]. In our networks, actors constitute the nodes, while flows of goods and money as well as shared institutions constitute the directional ties (for similar approaches in related areas of research see [29,60]). We descriptively analyzed the networks of flows and institutions among actors in order to shed light on telecoupled land governance [35]. The network lens makes it possible to disentangle the flows that diverse actor types exchange across levels, in addition to the institutions they share.

## 4. Results

### 4.1. Telecoupled Land Governance and Land Competition

Developing countries provide important commodities, biodiversity, and carbon sinks to the world. North-eastern Madagascar, in particular, satisfies demands from distant countries for agricultural goods and mining products. Apart from these tangible commodities, the region contributes significantly to global natural wealth, environmental knowledge, and climate change mitigation through its important biodiversity conservation areas and tropical forests. For our study villages, focus group interviews provided us with information about the LUs related to these different demands. According to our hypothesis that a combination of these demands would increase land competition, we classified them into two domains: the economic domain and the environmental domain, respectively, according to LU involving production or conservation [12]. The economic domain includes the production/trade of cash crops like clove and vanilla as well as the mining/trade of quartz crystals (semi-precious stone), whereas the environmental domain includes the stewardship of cash crops such as silk, the carbon credits market, and biodiversity conservation demands.

Figure 2 provides an overview of the three networks (flows of goods, flows of money, shared institutions) among these actors, for both study villages. Nodes represent the actors involved in the land governance network, with color codes distinguishing actors from three different sectors (public, private, or voluntary sectors). Further, the nodes are positioned in each network graph in Figure 2 according to the level of action they belong to, from the village level (bottom of each graph) to the international level (top); and from the economic domain (left) to the environmental domain (right). The network ties between the nodes represent the flows of goods (top two graphs of Figure 2), money (middle graphs), and shared institutions that the actors adhere to (bottom graphs). For the networks of goods and money flows, the ties convey an exchange of goods or money between the actors, once or more, as assessed in the survey and in the secondary data. In the institutional networks, two actors are linked by a tie if they share at least one institution.

Based on Figure 2, we can look for two different elements in these networks that may indicate the presence or absence of land competition under telecoupling. Firstly, in order to capture land competition, we can study the connections between the environmental and the economic domains. We can also examine at which level, and through which flows, the two domains are related. Furthermore, we can consider the sector to which the actors in each domain primarily belong. Secondly, in order to capture potential telecoupling and regulation thereof, we can assess the presence of distant actors. We can also explore which flows link these distant actors to the village level, and whether they link to the village level directly or via other levels. Additionally, we can analyze the network of institutions involving distant and local actors.

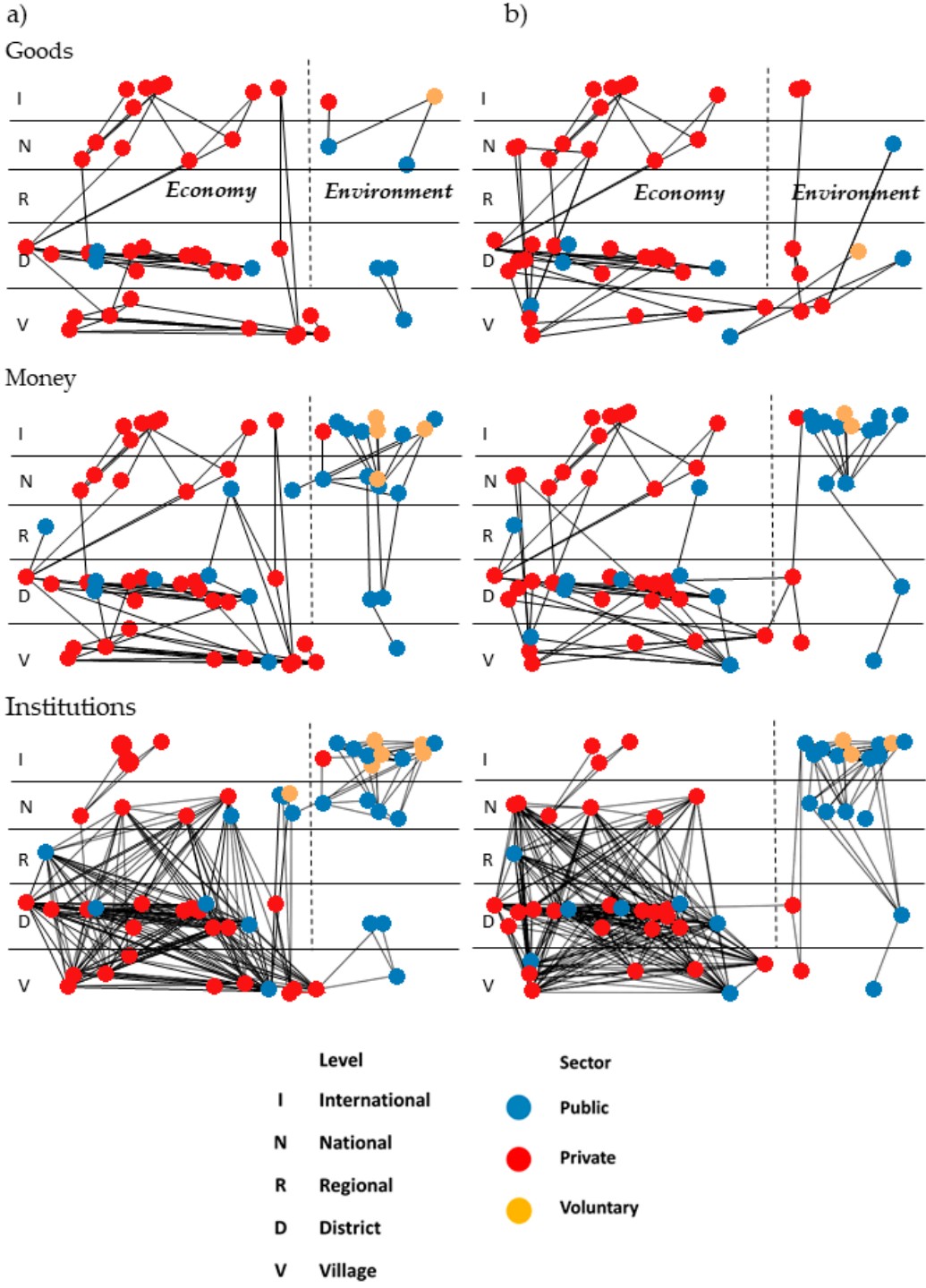

**Figure 2.** Network graphs representing interactions in terms of goods, money, information and institutions among the actors involved in land governance in the two case study areas of (**a**) Morafeno; and (**b**) Mahalevona. Variance in vertical positions of the nodes indicates the level to which the actors belong: international, national, regional, district, or village. The left/right of the dashed line represents the actors' domain: either economic, i.e., trade of cash crops and trade of quartz crystal (for Morafeno village only); or environmental, i.e., biodiversity conservation, silk network (for Mahalevona village only), and the carbon market (for Morafeno village only).

We begin by focusing on the potential interdependencies between both domains: economy and environment. While land competition is likely driven by increasing demands from both domains, here we analyze whether actors from both domains exhibit interdependencies through shared flows.

If actors from both domains do not interact in terms of flows or do not share institutions, they are likely unaware of each other's activities and strategies, which can increase local-level competition over land. Figure 2 reveals that there are very few flows and institutions that cross the vertical line, connecting actors from the economic and environmental domains. The few exceptions where exchanges between the development and conservation sectors occur are instances of LU practices that seek to integrate use and protection. In both villages, farmers are the main link between actors in the economic and environmental domains. Relatedly, it is at the village level that competition manifests between the domains. Nevertheless, the situation differs from one village to the other. Regarding the networks relevant to Morafeno village (left-hand graphs [a] in Figure 2), actors linking the domains include the "World Bank", from the economic domain, and the "Madagascar Biodiversity Fund" (FAPBM/MFB), from the environmental domain. Both are involved in the conservation network, exchanging flows of money for conservation funding. In comparison with Morafeno, the networks relevant to Mahalevona village (right-hand graphs [b] in Figure 2) feature more actors linking the environmental and economic domains. Two actors from the environmental domain—i.e., "Antongil conservation NGO (NGO Antongil)" and the park manager "Madagascar National Park (MNP Maroa.)" in Maroantsetra—are linked to an actor from the economic domain, i.e., the "rural commune of Mahalevona". A flow of goods occurred via donation of materials, thus linking the domains. Additionally, the actor "Sepali NGO" in Maroantsetra, from the environmental domain, generated a flow of money as tax payments to the "tax office in Maroantsetra (Perception_Maroa)", from the economic domain, for their silkworm trade. Further, the "Sepali NGO" (Maroantsetra; environmental domain) and the "Society Ramanandraibe Maroantsetra (Ramanandraibe Co.)" (economic domain) shared the same institution based on producing and/or transforming products certified Organic farming (AB France). In comparison to Mahalevona, the networks relevant to Morafeno village display very few actors linking domains. In sum, only a handful of actors bridge the two domains, such that there are few flows or shared institutions between them. They likely do not have the same goals in terms of LU, making the task of balancing LU planning harder and land competition more likely.

The actors from different domains (economic vs. environmental) who are placing demands on land also belong to different sectors. In Figure 2, actors in the private sector are indicated by red nodes. Private actors outweigh all others in the economic domain, whereas public-sector actors, indicated by blue nodes, constitute the majority in the environmental domain. Figure 3 shows that of the 38 actors in the environmental domain for both villages, 66.7% are from the public sector for Morafeno village and 70% are from the public sector for Mahalevona village. Environmental-domain actors from the private and voluntary sectors constitute 33.4% for Morafeno village and 30% for Mahalevona village. By contrast, in the economic domain (N = 131 for both villages), 76.6% of actors are from the private sector for Morafeno village and 85.1% are from the private sector for Mahalevona village, whereas 24.4% of actors are from the public or voluntary sector for Morafeno village and 14.9% are from the public or voluntary sector for Mahalevona village. Figures 2 and 3 highlight the dominance—at least numerically—of private sector actors in the economic domain and public sector actors in the environmental domain. Commodity trade actors (e.g., intermediaries) are most prominent among the private sector actors in the economic domain, while the main actors managing the protected areas—CBNRM and the REDD+ projects—are the most prominent public sector actors in the environmental domain. These actors pursue different aims and adopt different strategies. The state of Madagascar controls the public sector, which aims at providing a public service. The private actors, in turn, control the private sector and aim at making personal profits. These different aims of the two different sectors illustrate that land competition in the villages also involves competition over public versus private LU purposes (Supplementary Materials).

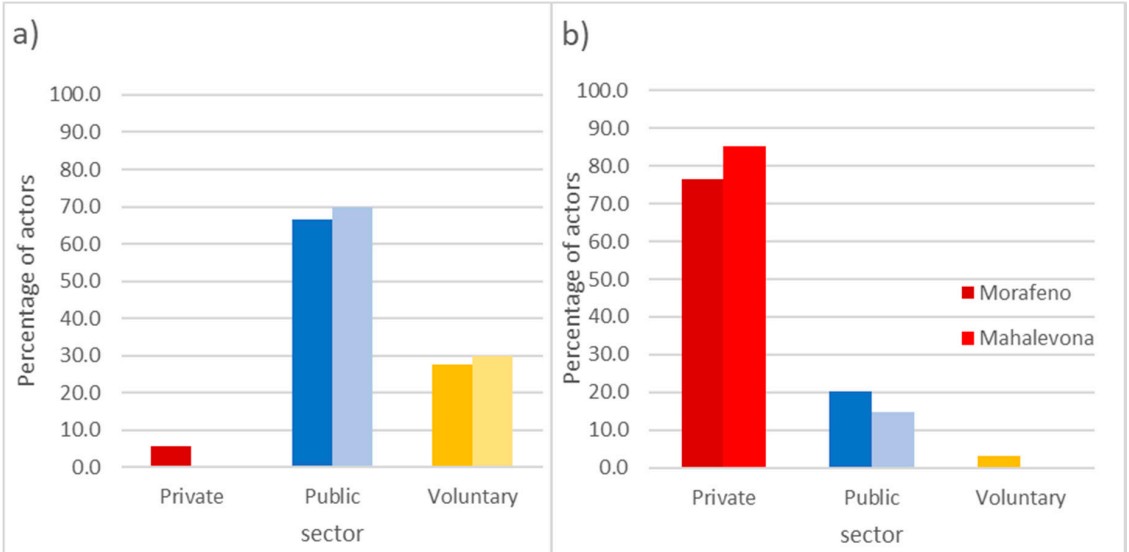

**Figure 3.** Distribution of the distant actors from the private, public, and voluntary sectors in the two domains in Morafeno (darker shading) and Mahalevona (lighter shading) villages: (**a**) distant actors from the environmental domain (N = 38), and (**b**) distant actors from the economic domain (N = 131).

Having analyzed the potential for land competition between both domains, we now turn to the question of levels of telecoupling. We focus on the position of international actors and their relations to the rest of the network in terms of flows of goods, money, and shared institutions. Figure 2 reveals that most networks involve international actors, both with respect to the economic domain and the environmental domain. More specifically, the flows of goods and money between international-level actors and village-level actors point to potentially telecoupled situations as well as competing demands on local land. In the case of Morafeno village, seven different international actors are involved in flows of goods for the economic domain. Whereas most of these flows of goods are linked to the local level via the national level, one international actor—a mining company headquartered in China—is also directly linked to the district level. The situation regarding flows of money is similar to that of flows of goods with respect to Morafeno village, at least for the economic domain. In the case of Mahalevona village, a similar pattern appears. Here, too, in terms of flows of goods and money, several international actors are linked to the local level via national actors.

The economic domain comprises cash crops, such as clove and vanilla, as well as quartz crystals, all representing export commodities. The demands for these products come from distant consumers. Their monetary values motivate local farmers to cultivate more cash crops and encourage miners to expand their mining activities. However, the recent creation of protected areas and the establishment of community-based natural resource management have rendered the remaining massive swathes of rainforest legally inaccessible to the farmers and miners. As a result, competition has increased over the remaining accessible land outside the restricted areas. The telecoupled context means that the rising demands of increasing numbers of distant consumers can continue to intensify such claims and competition over local land. At the same time, the shared institutions that several actors adhere to indicate the amount and type of telecoupling. An extremely dense network of shared institutions links relevant local-level actors to national-level actors (economic domain in Figure 2). However, international-level actors involved in the flows of goods and money tend either not to share the same institutions or only to share institutions with one national-level actor (e.g., conservation funding under trust fund). Overall, in the economic domain, the relative importance of international actors is not reflected in the institutional network. In other words, while international actors are strongly linked to the local level via exchanges of goods and money, these exchanges are seldom embedded in shared institutions.

However, a different picture emerges in the environmental domain: very few actors are involved in flows of goods, instead, flows of finances are most important. Further, most actors operate at the international and/or national levels, and have very little involvement with local actors. This is true for both of our case study villages. For example, global development and conservation initiatives led to the creation of both the protected areas in our study region—i.e., Masoala Natural Park and Makira Natural Park—as well as the community-based natural resource management system around the parks. The funding for these conservation activities stems from international actors, namely, international conservation organizations and/or foreign governments. Similarly, the buyers of carbon credits (e.g., Microsoft, a private-sector economic actor), sold within the REDD+ project in Makira Natural Park, are international-level actors. The network of shared institutions in the environmental domain strongly reflects the network of financial flows: a dense network between international and national actors, and almost no actor-to-actor connections at the local level. Yet related impacts at the local level can have important consequences on the LU of farmers. For example, the Protected Area Code or Code de Gestion des Aires Protégées—Loi n 2015—005 (among other regulations) prohibits local farmers and other stakeholders from extending their agricultural land and mining sites into delimited forests. In this way, distant actors create and implement institutions governing protected areas, and local actors must adhere to them. While farmers previously considered these forests as a reservoir of land for future cultivation especially by their descendants, today clearing new plots is legally forbidden. Inherited land is thus shrinking from generation to generation of Malagasy farmers. Finally, there is one important distinction between Morafeno and Mahalevona with respect to the environmental domain: in Morafeno, actors at the international, national, and local level do not adhere to the same institutions, even though finances flow between these levels. In Mahalevona, on the other hand, an institutional linkage exists (e.g., AB certification within the silk program), indicating that a shared institution covers the flow of money and goods.

In sum, given the presence of international actors in both domains, telecoupling is present with respect to economic and agricultural development in both villages. These international actors are almost never directly linked at the local level. Instead, flows occur via national-level actors that function as intermediaries. Notably, international and national actors appear more important in the environmental domain—in terms of their share of the overall number of actors per domain—than in the economic domain.

### 4.2. Distant Demands Reinforce Land Competition

Building on the above overview, we now zoom in on our case study villages to examine more closely how telecoupling reinforces local land competition. We further explore interactions between the two domains—economic and environmental—and their consequences for land competition. We consider farmers as part of the economic domain, and conservation actors as part of the environmental domain. Conservation actors support the farmers in diversifying their livelihoods via flows of goods such as clove seedlings and agricultural inputs.

Clove and vanilla plantations are highly telecoupled LUs in the Morafeno and Mahalevona villages, as they are strongly connected to international actors (Figures 4 and 5). The trade networks of clove and vanilla are dense compared with the networks of the crystal trade, biodiversity conservation, and carbon credit markets, as illustrated by the many ties of goods and money between actors at each level (Figures 4a and 5a). The same six international actors are linked to both villages via district-level actors and national-level actors. This relatively high number of international actors illustrates the significance of these LUs in terms of telecoupling compared with other LUs.

Further, Figures 4a and 5a show that the majority of actors operate at the district level, due to the high number of clove and vanilla collectors residing in the district capital of Maroantsetra. The flows of goods and money are most dense at the village level, spreading out as they travel via intermediaries towards the international level (Figure 2). Our analysis reveals that the importers of cash crops produced in Morafeno and in Mahalevona are located in India (Importers), Singapore

(Importers, Sivanil), France (Touton SA), Germany (Symrise AG), and the USA (Importers). These firms import spices to their countries from Malagasy district- or national-level exporters including the "Ramanandraibe company" (Ramanandraibe Co.) in Tamatave, the "Symrise company" (Symrise Ant.) in Antalaha, the "VanilleMad company" (Vanillemad) in Sambava, as well as "Trade Mark's company" (Trade Mark's) in Tamatave, and "Origines sarl" (Origines sarl) in Antananarivo. These firms buy vanilla and clove from individual national or district collectors (collectors) as well as one local farmer cooperative, associated with Symrise company in Antalaha (AGM). These collectors, in turn, buy clove and vanilla from other collectors and intermediaries. The intermediaries travel to Morafeno village and Mahalevona village to collect the commodities directly from the smallholder farmers. Some of the intermediaries cross district boundaries, as in the case of Mahalevona village, as they transport the commodities to the town of Antalaha. In most cases, the collectors provide advance payment to the intermediaries, who use it to buy the commodities after deducting their own commission. Collectors may also advance money to other intermediaries to gather the products.

Most of the actors present in the goods network are also present in the money network. Still, there are exceptions. Actors found solely in the money network are those who collect taxes, including the "rural communes of Morafeno and Mahalevona" (Rural commune), the "region of Analanjirofo" (Region Analanjirofo), the "perception of Maroantsetra" (Perception_Maroa), and traditional leaders such as the "association of elders in Morafeno" (Elders).

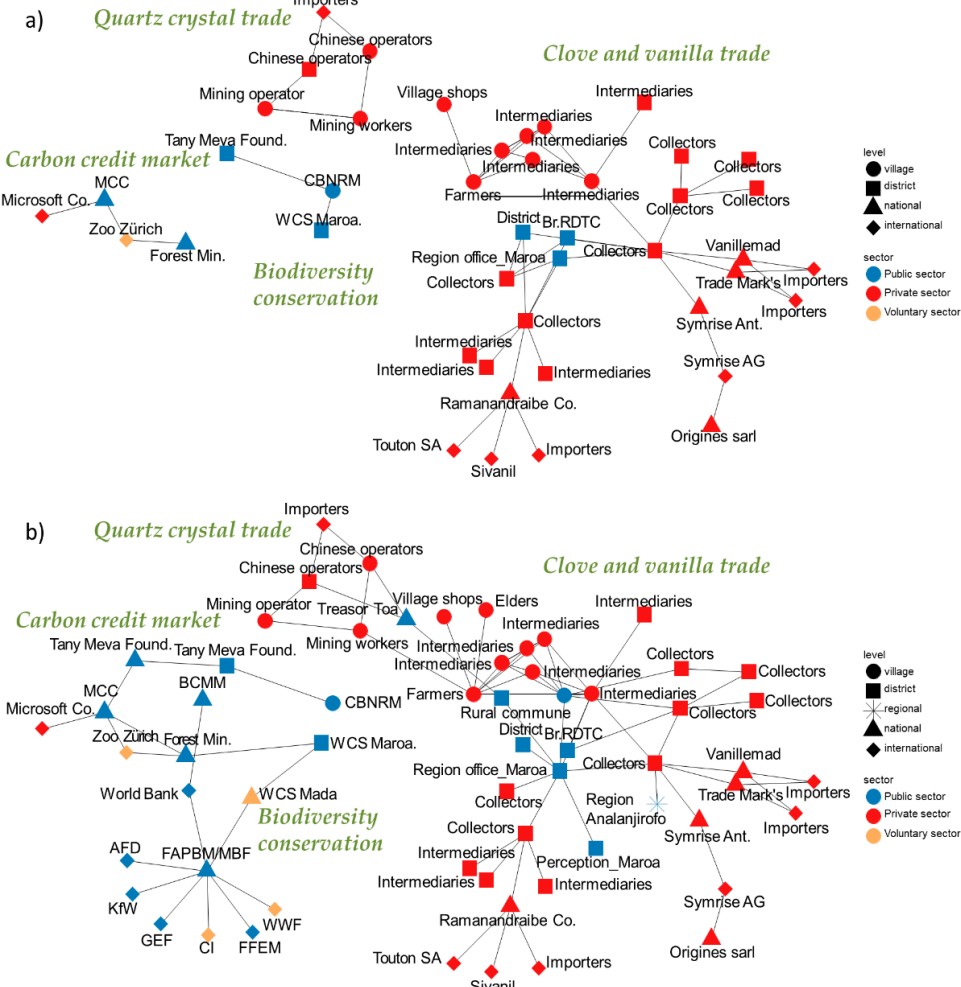

**Figure 4.** Network graphs of (**a**) flows of goods and (**b**) flows of money in Morafeno village representing the two domains: economic (e.g., trade in clove, vanilla, and quartz), and environmental (e.g., biodiversity conservation, carbon credit market).

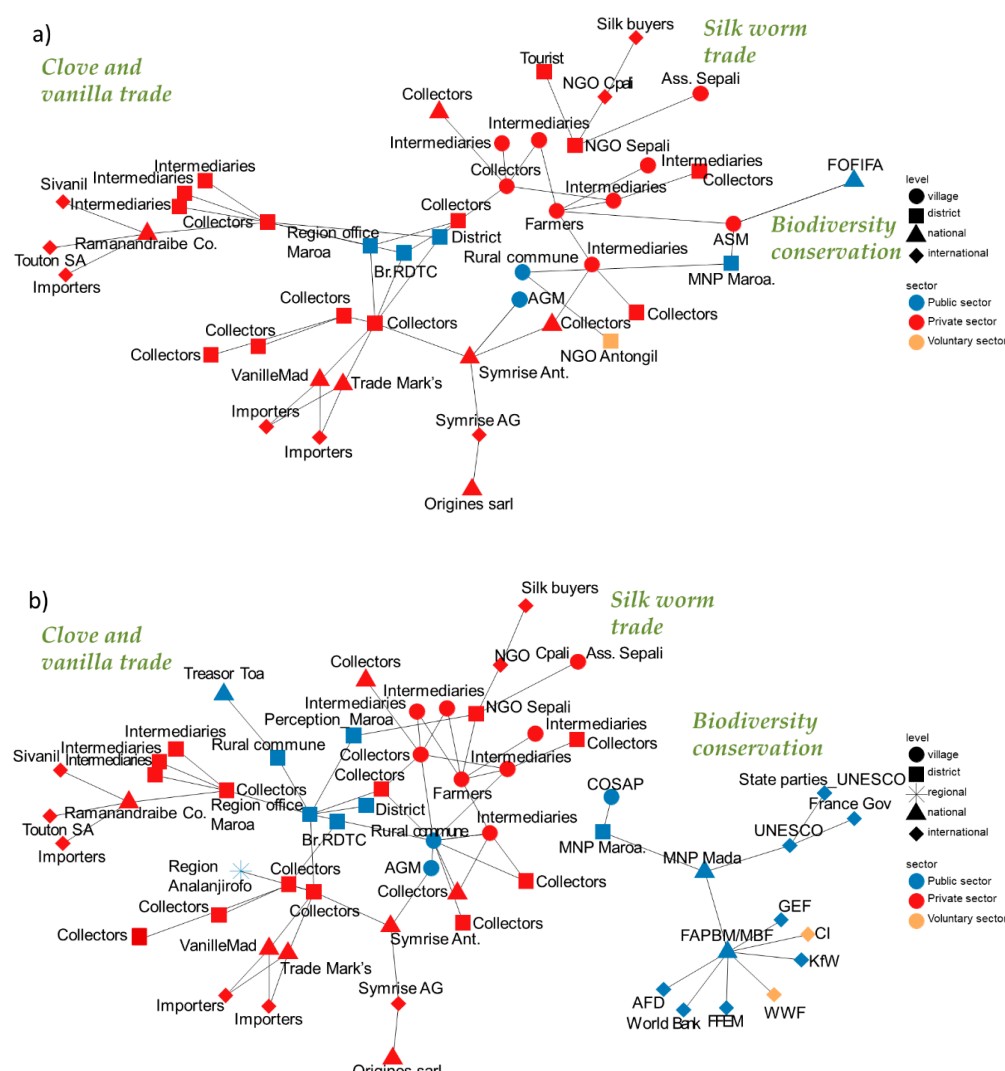

**Figure 5.** Network graphs of (**a**) flows of goods and (**b**) flows of money in Mahalevona village representing the two domains: economic (e.g., trade in clove, vanilla, and quartz), and environmental (e.g., biodiversity conservation, silkworm trade).

As described earlier, protected areas are another telecoupled LU present in both villages. Again, the biodiversity conservation networks differ from the economic networks insofar as goods are rarely traded in conservation, but money flows are very important (Figure 2). Figures 4 and 5, depicting conservation-related nodes and ties relevant to the two villages, show the actors involved in biodiversity conservation in Morafeno and Mahalevona villages and their interactions in terms of goods and money. While actors at the regional level are absent, there are many international-level actors.

The nodes and ties present in the conservation funding network (Figures 4b and 5b) point to relatively direct links from the international level to lower levels. More precisely, flows of money converge from international conservation donor actors to a few recipient actors at the national level. Indeed, north-eastern Madagascar is a meeting point of some of the world's most well-known global conservation funders. Our data point to the following key conservation actors: the "World Wildlife Fund" (WWF), "French Development Agency" (AFD), "French Global Environment Facility" (FFEM), "KfW-Bank aus Verantwortung" (KfW), "Global Environment Facility" (GEF), "Conservation International" (CI), "United Nations Educational, Scientific and Cultural Organization" (UNESCO), and the French Government (France Gov). These organizations have funded the creation and ongoing management of the Masoala National Park and Makira Natural Park in response to global conservation goals. The national foundation "Madagascar Biodiversity Fund" (FAPBM/MFB) gathers and manages

the funds. In our study area, "Madagascar Biodiversity Fund" provides funds to the national managers of the protected areas: "Madagascar National Parks" (MNP Mada) and "Wildlife Conservation Society" (WCS Mada) Madagascar. MNP receives further funding from the "Zoo of Zürich" (Zoo Zürich) in Switzerland, an international-level actor, following their agreement on the creation of a greenhouse exhibit called the "Petit Masoala" in the Zoo of Zürich. In our interview, Zoo Zürich stated that this agreement specifically establishes conservation and development around the national park as the focus of its allocated funds. The park managers "Madagascar National Parks (MNP) Maroantsetra" (MNP Maroa.) and "Wildlife Conservation Society (WCS) Maroantsetra" (WCS Maroa.) are key district-level organizations. To realize their aim of conservation driven by global initiatives and funded by international-level actors, both organizations collaborate directly with the local population at the village level. Our survey confirms that they provide local farmers with clove seedlings and agricultural inputs for cash-crop production, in an effort to help them diversify and improve their livelihoods. Their donations usually flow through "CBNRM Morafeno" (CBNRM) and the local association "Mutual Help Association" (ASM) in Mahalevona village. Further, MNP Maroantsetra established a committee, "Local Park Committee" (COSAP), in Mahalevona village to prevent violation of the park boundary and to support ecological monitoring of the park. Moreover, it provides materials such as solar panels to the "rural commune of Mahalevona" (Rural commune).

The two subdomains and their related LUs—i.e., clove/vanilla cultivation and biodiversity conservation—compete with each other for land. Their telecoupled status reinforces land competition. On the one hand, the farmers view the region's forests as a reservoir of land for expansion of cultivation as their family grows. Further, rising vanilla and clove prices motivate farmers to increase the size of their cash-crop plantations. On the other hand, the region's forests and surrounding areas are now officially off limits for agricultural expansion. The forests are classified as protected areas and any infraction is punishable by law. As in the case of the clove/vanilla trade, international demands lie behind the drive for biodiversity conservation. Distant actors shape and profit from both LUs. Yet there is very little interaction, within or across levels, between actors involved in the clove/vanilla trade (economic domain) and those involved in biodiversity conservation (environmental domain). With coordination between the actors lacking, the respective LUs compete based on differing claims to the same land.

Nevertheless, several examples of direct actor-to-actor interactions between the two sub-domains exist, which point to ways or efforts to balance the divergent claims on local land. One example is the support that conservation actors provide to farmers who are members of local conservation associations, in an attempt to improve the farmers' livelihoods. In the case of Morafeno, the conservation actor "Wildlife Conservation Society Maroantsetra (WCS Maroa.)" provides cash-crop seedlings and agricultural inputs to members of the local association "CBNRM in Morafeno (CBNRM)" village. In the case of Mahalevona, the conservation actor "Madagascar National Park in Maroantsetra (MNP Maroa.)" also provided seedlings and other inputs to the local farmer association "ASM Mahalevona (ASM)" during its launch. In the meantime, "ASM Mahalevona (ASM)" has become self-sustaining, and provides various tree seedlings to farmers at an affordable price. In this way, with the help of conservation actors, farmers were able to plant more cash crops, producing goods for international markets and benefiting from better prices.

However, providing farmers with means to cultivate more cash crops may encourage them to expand their cash cropping into forests. On the one hand, the trade of cash crops is presently flourishing in north-eastern Madagascar. Current clove and vanilla prices enable farmers to earn more income. This incentivizes young and old farmers alike, and even non-farmers, to get involved in the business of clove and vanilla production. On the other hand, conservation actors assume that cash-crop plantations will provide alternative livelihoods for farmers who have lost land access to protected areas and depend on shifting cultivation for subsistence rice production. The latter LU system has been frequently blamed for deforestation in the region. However, since cash cropping is highly profitable, farmers seek to enlarge their cropping areas and non-farmers seek new plots.

With cultivable land now scarce, the only opportunity for expansion would require clearing forest areas that are currently entirely protected. Indeed, the type of support provided by conservation actors to local farmers may actually be undermining conservation efforts in the long term. Further, considering the telecoupling situation, the absence of interaction or coordination between actors from the economic and environmental domains at other levels may aggravate this land competition between cash-crop plantations and biodiversity conservation.

### 4.3. Distant Institutions Reinforce Land Competition

With this understanding of the local impacts of telecoupled demands, we now turn our attention to the telecoupled institutions governing local LU decisions in Morafeno and Mahalevona villages with respect to the carbon credit market and the silk market (Figure 2). Empirical data reveals that land users base their decisions related to land governance on distant institutions regulating each of these sub-domains. Consequently, in these cases, distant and local actors share institutions that govern their exchanges of flows.

Distant institutions regulate the market for carbon credits linked to the Makira Natural Park in Morafeno and the silk market in Mahalevona village. On the one hand, the Makira Natural Park is linked to a REDD+ project that was launched in response to the Durban Vision Madagascar. The "avoided deforestation" sold through the REDD+ project complies with the Verified Carbon Standard (VCS) and the Climate, Community and Biodiversity (CCB) standards. These standards, created and implemented by actors at the international level (Verra), certify the emission reductions for buyers (individuals and companies) seeking to offset their carbon emissions. The buyers in our study area are private-sector economic actors such as "Microsoft (Microsoft Co.)" and "Zoo of Zürich (Zoo Zürich)". The standards are meant to ensure the "full and effective participation" of all stakeholders. However, village-level organizational actors involved in the REDD+ project do not appear to adhere to the standards, as deforestation is visibly ongoing. First, our survey results indicate that local farmers are unaware of the specific requirements of the international standards. Second, although entitled to part of the carbon funds through the benefit-sharing agreement in the REDD+ project documents, the farmers do not consider the current level of compensation to be sufficient.

Regarding the silk market, on the other hand, the France-based institution "Organic farming (AB France)" is responsible for certifying some of the local silk products as organic. This scheme certifies production that is respectful of the environment (regarding natural life cycles) and avoids use of chemical inputs on the farm. The farmers, members of a village-level silkworm-breeding association, must comply with these regulations when breeding the silkworms. Further, in accordance with the rules, the district-level "NGO Sepali" must avoid use of chemical inputs when transforming the farmers' product into raw silk. Our survey indicates that both the farmers and the NGO do, indeed, comply with the certification rules. In this case, the local NGO strongly enforces the regulations at the village level. In addition, actors at the international level belonging to the silkworm value chain uphold the distant regulation. Through this particular institution, farmers appear to benefit from the improvement of their livelihoods.

These telecoupled institutions can increase land competition if actors at different levels do not adhere to them. In the example of carbon credits in Morafeno village, the farmers at the village level do not adhere to the certification standards. They do not perceive the compensation as sufficient for their loss of forest access, which has substantial implications for their livelihoods (undernutrition). As a result, they are unwilling to give up their forest-related livelihoods, despite the existence of the REDD+ project and the protected area. They continue to practice shifting cultivation to produce rice, which they were supposed to abandon in exchange for REDD+ compensation. Further, with the increasing prices for cash crops, farmers are tempted to expand their plantations into forests. In this way, land competition is reinforced on the ground because distant institutions fail to incorporate actors at every level and to coordinate between the two domains.

Yet the opposite is illustrated by the example of the silk program and its AB certification. Every actor along the silkworm value chain adheres to the AB regulation, and enforcement provided by the NGO Sepali at the village level is strong. The farmers benefit from the silk market and comply with the regulations because it improves their livelihoods. Silkworm breeding, usually occurring on existing agroforestry plots, does not compete with the forests or other LUs. Its farming practices and possible expansion are kept under control by the shared institution of AB certification, and the rules are followed by actors at every level. Further, the two domains are brought together: farmers, representing the economic domain; and the local breeders' association and NGOs, representing the environmental domain. Incorporating the two domains and actors from each level while providing benefits to farmers on the ground helps to prevent land competition.

## 5. Discussion

With respect to land competition in our case study sites, we find little evidence of interactions, coordination or negotiation, between actors in the economic versus environmental domains who are linked to the villages of Morafeno and Mahalevona. Conservation activities appear largely disconnected from economic activities, until they reach the local level, where they target the same land and land users. Furthermore, the lack of interaction between actors/flows in either domain appears to contribute to a case of Jevons paradox [61,62]: the increased accessibility of cash-crop seedlings, facilitated by higher-level conservation actors, enhances farmers' LU efficiency. Yet this also feeds pressure to expand cash-crop plantations, as external demands rise. While conservation is important to maintain ecosystem services and mitigate biodiversity loss [63] and can be structured to support development [64], conservation efforts appear to be mainly top-down in Madagascar, with local communities largely excluded from decision-making [65]. Conservation goals funded by distant actors are not fully met [66,67] and local communities continue to live in extreme poverty [67]. Additionally, the local people in our villages readily seek income alternatives to conservation, as demonstrated by LU related to cash cropping, mining booms, and general land competition [68].

At the same time, study results from other countries show that protected areas can benefit local populations. Positive impacts on household income have been found around protected areas in an Integrated Conservation and Development Projects (ICDP) in the Brazilian Amazon [69]. Benefits for livelihood diversity have been identified regarding marine protected areas in Indonesia [63]. These and other research findings on socioeconomic improvements linked to sustainable-use protected areas [16] highlight the potential of reconciling the aims of conservation and development. Harmonization appears to require sustainable resource use, co-management of protected areas by conservation agencies in collaboration with communities [16], and properly addressing local needs, expectations, and attitudes towards conservation [70]. Other keys include local ownership and social-economic integration [71–73] as well as engagement at higher levels of governance [74]. Appropriate policies would prevent increased agricultural production at the expense of local forests while ensuring people's food and livelihood security [2]. Recognizing the agency of local actors, as agents of change, is a prerequisite to foster sustainable development in the biodiversity hotspot of north-eastern Madagascar [75].

Besides lack of coordination, actors belonging to different sectors are dominant in the two different domains, which can reinforce land competition. Some observers [27] highlight the dominance of the private sector (traders/companies) over the public sector (state) in current land governance. The private sector could provide more support to the public sector in its tasks and roles, reinforcing the central position of the latter in the land-governance system. In Indonesia, the private sector contributes significantly to REDD+ governance through private markets and finances [31]. However, other observers [76] claim that competition can result between governments, NGOs, and corporations. With this in mind, we found that regulation of the commodities trade and conservation activities in Madagascar officially remain the responsibility of the public sector, i.e., the local and national Malagasy government. This regulating role is smaller than in times past [30], as the vanilla market

was a state-run monopoly prior to 1993 [77]. Today, states appear to be reasserting their regulatory power, such as in the case of forest and agricultural certification in Indonesia [78] or "land grabbing" governance [79] globally. Collaboration between public and private sectors can be effective under sustainable institutional contexts [76,80]. To achieve sustainability in our study context, more in-depth research on key actors appears necessary [59,81].

In terms of telecoupling, our results show that distant flows and institutions carry claims from different domains and exert pressure on the land systems of Morafeno and Mahalevona villages. Both domains—economic and environmental—convey distant influences. Our results represent a case of international nongovernmental organizations (INGOs) progressively introducing a global conservation agenda in the global South [82]. This promotion of conservation via the creation of protected areas, community-based conservation, and designated carbon sinks is due to global loss of taxonomy/functional biodiversity [16] and the threats posed by climate change worldwide [83]. Notably, these conservation activities critically depend on funding from international organizations. On the ground in north-eastern Madagascar, this involvement of distant actors in conservation, as currently structured and implemented, appears to compound land competition. Long-range pressures manifest among village-level land users, who are caught between conservation and economic development strategies. On the conservation side, the REDD+ project and protected areas in our study area do not appear to be fulfilling their aims of improving livelihood outcomes for local beneficiaries, as they apparently provide insufficient financial compensation for the foregone benefits of forest land [68]. Dissatisfaction with compensation has been shown elsewhere to result in ineffective conservation outcomes, such as deforestation and forest degradation [16]. In the meantime, local land users seize other available opportunities in the economic domain, many of which threaten natural habitats. In our case study areas, many are turning to artisanal mining of quartz. However, many also seek to expand their cash-crop plantations, especially for vanilla production, putting pressure on the remaining forests. This undermines the professed goals of fostering intensification of LU and abandonment of shifting cultivation.

Using the telecoupling concept as a lens to investigate the different actors linked to local LUC enabled us to better understand land governance in north-eastern Madagascar, and its implications for sustainable development. We analyzed LU decision-making networks in a systematic, detailed manner, visualizing social networks and thus revealing distant influences and their local effects on LU. Telecoupled land competition threatens to undermine sustainable human–environmental interactions over the long term [80]. Bringing together actors from the environmental and economic domains and encouraging them to better coordinate land governance could enable more sustainable development in the region.

As we took local LU changes as the starting point for our snowballing survey of actors, the local system naturally received a deeper analysis than distant systems. Our study is limited regarding insights into telecoupled international-level LU changes: We did not determine what effect these telecoupled sites in Madagascar have on other distant/local land systems. Further, we did not distinguish between links of goods versus money in terms of strength, and we did not distinguish between different types of institutions. Further research could investigate these issues.

## 6. Conclusions

Telecoupled land governance in north-eastern Madagascar reinforces land competition. Distant flows and institutions influence land users and other actors involved in land governance. The relevant actors broadly belong to two different domains: economic and environmental. They generally do not interact, and different sectors dominate either domain. More interdomain engagement, negotiation, and coordination is needed to reduce the land competition reinforced by telecoupling. These findings shed light on urgent land governance issues in Madagascar and similar developing countries. Solutions for land-related issues such as resource degradation and illicit trade flows must consider these distant influences. To shed light on many issues of land governance, we must move from place-based to

flow-based approaches. The current study makes progress towards operationalizing the concept of telecoupling. Flows and agents, proposed by Liu et al. [6] are at the heart of our study.

Finding a good balance between the environment and economic domains at the local level requires improved spatial planning of LU—involving not only local farmers, but also key actors from the two domains—and adaptation of national goals for agricultural commodity production. Making land governance more effective also requires in-depth study of the key agents of change among land governance actors. The sustainable development challenges in north-eastern Madagascar require collaboration between actors across scales and domains. Key actors with differing backgrounds and objectives must harmonize their interests to protect the island's globally relevant natural heritage and to bring economic prosperity to its inhabitants.

**Supplementary Materials:** The following are available online at http://www.mdpi.com/2071-1050/11/3/851/s1, Data S1: coord_Mahfv.csv, Data S1: coord_Morfv.csv, Data S1: data_attributes_actors_all.csv, Data S1: flowsMah.csv, Data S1: flowsMor.csv, Data S1: inst.attributes.csv, Data S1: xycoordsMah.csv, Data S1: xycoordsMor.csv, Code S1: Rscript.R file, Figure S1: legend, Data S2: Distribution_distant_actors_data.xlsx, Survey S3: Actor_survey.docx.

**Author Contributions:** O.R.A. conceived the approach, conducted the fieldwork and case study, and drafted and finalized the manuscript. O.R.A. and F.M. collaborated on the R codes for the network graphs. O.R.A. and J.G.Z. collaborated on the case study map. All authors contributed to discussion of the concept, methods, and results of the study, and commented on draft versions.

**Funding:** This work was supported by the Centre for Development and Environment (CDE) and the Institute of Geography, University of Bern, Switzerland; as well as the Swiss Programme for Research on Global Issues for Development, funded by the Swiss National Science Foundation (SDSN) and the Swiss Agency for Development and Cooperation (SDC) [grant number 400440 152167].

**Acknowledgments:** The research was carried out as part of the project titled "Managing telecoupled landscapes for the sustainable provision of ecosystem services and poverty alleviation" (Project No. 152167) in Madagascar in collaboration with, and with extensive support from, the Department of Forestry and Environment at the School of Agronomy in the University of Antananarivo, Madagascar. We thank all the surveyed and interviewed actors for their precious time spent answering our questions. We also thank all the village authorities in the two villages for their cooperation. This work would not have been possible without our main research guide, Paul Clément Harimalala, and all the other local guides involved. We thank Anu Lannen for English editing.

**Conflicts of Interest:** The authors declare no conflict of interest.

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
