# Peer review of "Land Competition under Telecoupling: Distant Actors’ Environmental versus Economic Claims on Land in North-Eastern Madagascar"

_sustainability, doi:10.3390/su11030851_

Round 1

Reviewer 1 Report

General comments

Competing land-uses and land-users determine the mode of utilizing lands, and potentially impacting environmental sustainability and the provision of ecosystem services. This study presents a framework for communications and interactions between the major local and distant actors, with reflections on both economic and environmental issues. The authors shortly mention (in the introduction section) the bilateral soy trade between Brazil and China, and how it impacts environmental sustainability and ecosystem services in the two countries. At least in environmental terms, this trade has led to a one winner side (China) and one looser side (Brazil). Many types of these relations occur around the world, where a strong-and-rich external actor exploits resources that are located/hold in/by a weak-and-poor actor.  Here, the authors discuss a different set of (very complex) interactions, where environmental benefits (nature conservation) are mostly local (Madagascar), and yet, economic burdens (halted development) are also local. The study deals with interactions among several levels, including village, district, region, national, and international. This is a good example of competing needs/services/goods at local scale, where distant actors support both environmental conservation and international trade of agricultural and mining products. The authors clearly identify dominancy by the private section in the economic domain, vs. dominancy by the public sector in the environmental domain. One way or another, the most important aspect, as clearly identified by the authors (lines 153-154) is the mismatch between local resource wealth and ongoing human poverty. The topic described in this manuscript seems innovative and interesting. Yet, I wonder what are the practical implications of this study, and particularly, in terms of sustainable development for agricultural communities in less developed countries. Specifically, it seems that regulatory compensation mechanisms such as the REDD+ cannot provide the farmers with sufficient alternatives. At the same time, encouraging of climate smart agricultural practices (e.g., agroforestry) and agro-tourism might diversify and strengthen sources of income for these rural communities. I would ask the authors to insert a paragraph on this topic, at the end of the discussion section.

Fig. 2.

The Y axis (V, D, R, N, I) is clearly understood. Yet, it is not clear if there is a meaning to the X-axis, and if yes, what it is.

Fig. 3.

The figure is unclear. What are the dark blue vs. pale blue pertain for? What are the dark yellow vs. pale yellow pertain for?

Author Response

Response to Reviewer 1 Comments

Point 1: I wonder what are the practical implications of this study, and particularly, in terms of sustainable development for agricultural communities in less-developed countries.

Response 1: We acknowledge the concern of the reviewer about the practical implications of this study. The results suggests that, in order for the agricultural communities to experience sustainable development and to facilitate their land use decision-making under land competition and telecoupling, actors from different levels and sectors, including the farmers, need to interact. Practically, these interactions could either be through direct collaboration at the local level, e.g. joint land use planning between conservation, commercial trade, as well as governmental actors or through regulative land and land-use related policies at the national level. We explain this in the last paragraph of the conclusion section (lines 704-711).

Point 2: It seems that regulatory compensation mechanisms such as the REDD+ cannot provide the farmers with sufficient alternatives. At the same time, encouraging of climate smart agricultural practices (e.g., agroforestry) and agro-tourism might diversify and strengthen sources of income for these rural communities. I would ask the authors to insert a paragraph on this topic, at the end of the discussion section.

Response 2: We appreciate the suggestions of the reviewer in terms of alternative incomes for the farmers. However, in this specific context and the situation of land competition under telecoupling, the existing crops under agroforestry are already competing with biodiversity conservation and other LUs. The trade-offs we observe are at the level of land use systems (e.g. commercial crop production, shifting cultivation, and forest conservation), and not at the level of specific agricultural practices or individual crops. Introducing alternative land-based incomes might only increase the pressure on the forest, as they also provide incentives for expansion. Instead, REDD+ or other mechanisms would need to provide adequate financial compensation accounting for the real value of foregone land use. We amended our explanation to clarify this point. (lines 670-671)

Point 3: Fig. 2. The Y axis (V, D, R, N, I) is clearly understood. Yet, it is not clear if there is a meaning to the X-axis, and if yes, what it is.

Response 3: We thank the reviewer for indication of the need to be more explicit about the X-axis. It shows the division of the two domains (economic and environment). The economic domain network is composed of cash crops network and the quartz crystal network (for Morafeno village only). The biodiversity conservation network, the silk network (for Mahalevona village only) and the carbon market network (for Morafeno village only) constitute the environmental domain. This information was added to the caption of Fig. 2.

Point 4: Fig. 3. The figure is unclear. What are the dark blue vs. pale blue pertain for? What are the dark yellow vs. pale yellow pertain for?

Response 4: We have integrated the missing piece of information into the caption of Fig. 3: the nuanced shading for each colour distinguishes the two villages for the same category of sector: darker shading for Morafeno village and lighter shading for Mahalevona village. Distinguishing the sectors by colour is actually not necessary for this figure, but it allows to link the results displayed in this figure to the network graphs in Fig. 2.

Reviewer 2 Report

Research stand at very high level. Authors propose new and very interesting approach to studies on linking land use changes with actors of change. Introduction section in broadly context presents background of research. Study areas as well as materials and methods are presented in very clear way. Results section can be improved by adding the table or graph which show the connection between main land use changes identified in the first part of studies during focus group interviews and the number of different types of actors at different levels which drive the changes.

Discussion section can be improved by adding information of limitations and possible errors/mistakes of the study as well as directions of further studies in this field.

Good job Authors.

Author Response

Response to Reviewer 2 Comments

Point 1: Results section can be improved by adding the table or graph which show the connection between main land use changes identified in the first part of studies during focus group interviews and the number of different types of actors at different levels which drive the changes.

Response 1: We thank the reviewer for his/her suggestion to improve the results section. However, such a descriptive analysis would not reveal the drivers of land use. We started our empirical data collection from these LUCs to identify the first actors linked to LUs and LUCs, acknowledged by the village actors. An identification of the specific drivers of LUC among the actors identified would require an additional qualitative analysis of their agency, which exceeds the scope of this paper. A social network analysis alone could be misleading.

Point 2:Discussion section can be improved by adding information of limitations and possible errors/mistakes of the study as well as directions of further studies in this field.

Response 2: The requested information is already presented in the conclusion section. However, we think that the reviewer’s suggestion to move it to the discussion section makes sense and improves the readability of the paper. We have therefore taken up this proposal.